# Functional genome-wide siRNA screen identifies *KIAA0586* as mutated in Joubert syndrome

Susanne Roosing[1]*, Matan Hofree[2,3], Sehyun Kim[4†], Eric Scott[1], Brett Copeland[1], Marta Romani[5], Jennifer L Silhavy[1], Rasim O Rosti[1], Jana Schroth[1], Tommaso Mazza[5], Elide Miccinilli[5], Maha S Zaki[6], Kathryn J Swoboda[7], Joanne Milisa-Drautz[8], William B Dobyns[9], Mohamed A Mikati[10], Faruk İncecik[11], Matloob Azam[12], Renato Borgatti[13], Romina Romaniello[13], Rose-Mary Boustany[14,15], Carol L Clericuzio[16], Stefano D'Arrigo[17], Petter Strømme[18,19], Eugen Boltshauser[20], Franco Stanzial[21], Marisol Mirabelli-Badenier[22], Isabella Moroni[23], Enrico Bertini[24], Francesco Emma[25], Maja Steinlin[26], Friedhelm Hildebrandt[27], Colin A Johnson[28], Michael Freilinger[29], Keith K Vaux[1], Stacey B Gabriel[30], Pedro Aza-Blanc[31], Susanne Heynen-Genel[31], Trey Ideker[2,3], Brian D Dynlacht[4], Ji Eun Lee[32], Enza Maria Valente[5,33], Joon Kim[34], Joseph G Gleeson[1]*

[1]Laboratory for Pediatric Brain Disease, New York Genome Center, Howard Hughes Medical Institute, The Rockefeller University, New York, United States; [2]Department of Computer Science and Engineering, University of California, San Diego, San Diego, United States; [3]Department of Medicine, University of California, San Diego, San Diego, United States; [4]Department of Pathology and Cancer Institute, Smilow Research Center, New York University School of Medicine, New York, United States; [5]IRCCS Casa Sollievo della Sofferenza, Mendel Institute, San Giovanni Rotondo, Italy; [6]Clinical Genetics Department, Human Genetics and Genome Research Division, National Research Center, Cairo, Egypt; [7]Departments of Neurology and Pediatrics, University of Utah School of Medicine, Salt Lake City, United States; [8]Department of Pediatric Genetics, University of New Mexico, Albuquerque, United States; [9]Center for Integrative Brain Research, Seattle Children's Hospital, Seattle, United States; [10]Division of Pediatric Neurology, Department of Pediatrics, Duke Institute for Brain Sciences, Duke University Medical Center, Durham, United States; [11]Department of Pediatric Neurology, Cukurova University Medical Faculty, Balcali, Turkey; [12]Department of Pediatrics and Child Neurology, Wah Medical College, Wah Cantt, Pakistan; [13]Neuropsychiatry and Neurorehabilitation Unit, Scientific Institute IRCCS Eugenio Medea, Bosisio Parini, Italy; [14]Departments of Pediatrics, Adolescent Medicine, American University of Beirut Medical Center, Beirut, Lebanon; [15]Departments of Biochemistry and Molecular Medicine, American University of Beirut Medical Center, Beirut, Lebanon; [16]Division of Genetics/Dysmorphology, Department Pediatrics, University of New Mexico, Albuquerque, United States; [17]Developmental Neurology Division, Fondazione IRCCS Istituto Neurologico Carlo Besta, Milan, Italy; [18]Women and Children's Division, Oslo University Hospital, Oslo, Norway; [19]Department of Medical Genetics, University of Oslo, Oslo, Norway; [20]Department of Pediatric Neurology, University Children's Hospital, Zurich, Switzerland; [21]Department of Pediatrics, Genetic Counselling Service, Regional Hospital of Bolzano, Bolzano, Italy; [22]Child Neuropsychiatry Unit, Department of Neurosciences and Rehabilitation, Istituto G. Gaslini, Genoa, Italy; [23]Unit of Child Neurology,

*For correspondence: sroosing@ rockefeller.edu (SR); jogleeson@ rockefeller.edu (JGG)

Present address: †Laboratory for Pediatric Brain Disease, New York Genome Center, Howard Hughes Medical Institute, The Rockefeller University, New York, United States

Fondazione IRCCS Istituto Neurologico Carlo Besta, Milan, Italy; [24]Unit of Neuro-muscular and Neurodegenerative Disorders, Laboratory of Molecular Medicine, Bambino Gesù Children's Research Hospital, IRCCS, Rome, Italy; [25]Division of Nephrology and Dialysis, Bambino Gesù Children's Hospital, IRCCS, Rome, Italy; [26]University Children's Hospital, Berne, Switzerland; [27]Division of Nephrology, Department of Medicine, Boston Children's Hospital, Howard Hughes Medical Institute, Harvard Medical School, Boston, United States; [28]Section of Ophthalmology and Neurosciences, Wellcome Trust Brenner Building, Leeds Institute of Molecular Medicine, University of Leeds, St. James's University Hospital, Leeds, United Kingdom; [29]Neuropediatric group, Department of Paediatrics and Adolescent Medicine, Medical University Vienna, Vienna, Austria; [30]Broad Institute of Harvard and Massachusetts Institute of Technology, Cambridge, United States; [31]High Content Screening Systems, Sanford-Burnham Institute, La Jolla, United States; [32]Samsung Genome Institute, Department of Health Sciences and Technology, Samsung Advanced Institute of Health Sciences and Technology, Sungkyunkwan University, Seoul, Republic of Korea; [33]Section of Neurosciences, Department of Medicine and Surgery, University of Salerno, Salerno, Italy; [34]Korea Advanced Institute of Science and Technology, School of Medical Science and Engineering, Daejeon, Republic of Korea

**Abstract** Defective primary ciliogenesis or cilium stability forms the basis of human ciliopathies, including Joubert syndrome (JS), with defective cerebellar vermis development. We performed a high-content genome-wide small interfering RNA (siRNA) screen to identify genes regulating ciliogenesis as candidates for JS. We analyzed results with a supervised-learning approach, using SYSCILIA gold standard, Cildb3.0, a centriole siRNA screen and the GTex project, identifying 591 likely candidates. Intersection of this data with whole exome results from 145 individuals with unexplained JS identified six families with predominantly compound heterozygous mutations in *KIAA0586*. A c.428del base deletion in 0.1% of the general population was found in *trans* with a second mutation in an additional set of 9 of 163 unexplained JS patients. *KIAA0586* is an orthologue of chick *Talpid3*, required for ciliogenesis and Sonic hedgehog signaling. Our results uncover a relatively high frequency cause for JS and contribute a list of candidates for future gene discoveries in ciliopathies.

**eLife digest** Joubert syndrome is a rare disorder that affects the brain and causes physical, mental, and sometimes visual impairments. In individuals with this condition, two parts of the brain called the cerebellar vermis and the brainstem do not develop properly. This is thought to be due to defects in the development and maintenance of tiny hair-like structures called cilia, which are found on the surface of cells.

Currently, mutations in 25 different genes are known to be able to cause Joubert syndrome. However, these mutations only account for around 50% of the cases that have been studied, and the 'unexplained' cases suggest that mutations in other genes may also cause the disease.

Here, Roosing et al. used a technique called a 'genome-wide siRNA screen' to identify other genes regulating the formation of cilia that might also be connected with Joubert syndrome. This approach identified almost 600 candidate genes. The data from the screen were combined with gene sequence data from 145 individuals with unexplained Joubert syndrome. Roosing et al. found that individuals with Joubert syndrome from 15 different families had mutations in a gene called *KIAA0586*. In chickens and mice, this gene—known as *Talpid3*—is required for the formation of cilia.

Roosing et al.'s findings reveal a new gene that is involved in Joubert syndrome and also provides a list of candidate genes for future studies of other conditions caused by defects in the formation of cilia. The next challenges are to find out what causes the remaining unexplained cases of the disease and to understand what roles the genes identified in this study play in cilia.

## Introduction

A range of disorders from isolated organ defects like blindness or nephronophthisis to multi-system disorders like Joubert (JS), Bardet–Biedl, or Meckel–Gruber syndromes are correlated with mutations in genes involved in formation or stability of the primary cilium (*Goetz and Anderson, 2010*; *Waters and Beales, 2011*; *Brown and Witman, 2014*). JS is characterized by a distinctive midbrain–hindbrain malformation, named the 'molar tooth sign' on brain magnetic resonance imaging, and clinically by developmental delay, oculomotor apraxia and hypotonia. Currently, 25 genes are known to cause JS when mutated in a bi-allelic or X-linked fashion (*Akizu et al., 2014*; *Beck et al., 2014*; *Romani et al., 2014*). Most of the encoded proteins from these genes localize to the primary cilium or are involved in ciliary-related transport and commonly result in defective ciliation in patient cells or in animal models (*Singla et al., 2010*; *Valente et al., 2013*; *Akizu et al., 2014*). Importantly, still about half of cases studied by exome sequencing remain genetically unsolved, suggesting many as yet unidentified causes (*Akizu et al., 2014*).

Although traditional homozygosity mapping or exome sequencing has uncovered many genes for these conditions, these approaches may fall short for genes under strong selective pressure or for genes in which homozygous loss-of-function mutations are embryonic lethal. One approach to identify new human disease genes is to intersect cell biological, genomic, or protein interaction data in order to prioritize candidates for closer inspection. For instance, a protein interaction network derived from genes previously implicated in the ciliopathies identified mutations in *TCTN2* in JS patients (*Sang et al., 2011*). Similarly, comparing gene content from species with and without cilia led to identification of *BBS5* in Bardet–Biedl syndrome patients (*Li et al., 2004*).

There have been few systematic approaches towards characterization of genes required for ciliogenesis. A small interfering RNA (siRNA) screen of 7784 pharmacologically relevant genes identified 36 positive and 13 negative ciliogenesis modulators (*Kim et al., 2010*), and a study of 815 'kinome' genes identified 9 candidates affecting ciliary signaling (*Evangelista et al., 2008*), but neither study was genome-wide. A recent phylogenetic co-occurrence study identified 206 core cilia components (*Dey et al., 2015*), but no link with disease was shown. Given defective ciliogenesis in patient cells, we reasoned that a genome-wide siRNA screen to identify ciliogenesis factors could help prioritize candidates, especially for families in which traditional exome-sequencing approaches have not yet yielded a cause.

One of the caveats of screening for such genes is that ciliogenesis is intimately linked with mitosis (*Kim et al., 2011*; *Plotnikova et al., 2012*), and thus, genes arresting the cell cycle prior to ciliogenesis might be inadvertently flagged as affecting ciliogenesis. Recent live cell cycle imaging markers make it possible to separately flag cell cycle genes, which could greatly increase the specificity of ciliogenesis screens.

Our focus was to identify novel genes involved in JS, by applying a functional genomics approach, then intersecting the data with a cohort of unsolved exome-sequencing results from JS patients. We conducted a high-throughput genome-wide siRNA knockdown study for 18,045 human genes in a 'two-color' cell line engineered to report ciliary-localized EGFP and cells in G2/M phase using mCherry-tagged Geminin. A range of cellular features were measured for all genes, and compared with a positive and negative training set, resulting in a prioritized list of 591 ciliary candidates. This list was used to prioritize variants from 145 JS patients on whom exome sequencing had not revealed a cause. We identified deleterious variants *KIAA0586* in a total of 15 families. This gene was previously missed by exome sequencing, most likely due to a high-carrier frequency of a common allele in a predominantly compound heterozygous inheritance, thus, precluding a homozygosity mapping approach or filtering focused on rare variants. Together with a lethal phenotype in other species (*Bangs et al., 2011*), the data suggest that humans may have redundancy or compensation that preclude lethality or that the *KIAA0586* mutations only partially inactivate protein function. The results also support a cell-based screening approach to complement exome sequencing in human mutation identification.

## Results

### Generation of SEMG cell line

The ciliated stable cell line, human telomerase reverse transcriptase (hTERT)-retinal pigment epithelial 1 (RPE1) Smo-EGFP (*Kim et al., 2010*), in which Smoothened–tagged EGFP is stably integrated in the

polarized human RPE1 cells, reliably reports a single primary cilium upon serum withdrawal in 60–80% of cells. This line was stably transfected with mCherry-tagged Geminin (*Sakaue-Sawano et al., 2008*), a nuclear marker for S/G2/M cell cycle phases, to produce the Smo-EGFP-mCherry-Geminin/hTERT-RPE1 (SEMG) line, enabling differential analysis of ciliogenesis as a function of the cell cycle. Cells lacking a cilium (i.e., absent ciliary-localized EGFP fluorescence) were divided into those in G2/M phase (should normally not display a cilium) and those in G0/G1 (most should display a cilium; *Figure 1A,B*). The incorporation of mCherry-Geminin increased the specificity of the screen by filtering siRNAs leading to cell cycle arrest as the primary reason for absent cilia.

Using this approach, we first optimized seeding density, serum withdrawal conditions, and imaging parameters using a siRNA positive control for cilia (i.e., no known effect on the cell cycle but blocking ciliogenesis) of KIF3A, and for cell cycle (i.e., no direct effect on ciliogenesis but traps cells in G2/M phase of the cell cycle or the effect described above) of ACTR3 and CRNKL1, and verified reporters were robust (*Figure 1C*).

## Cell-based screen and validation of whole-genome siRNA data set

We conducted a high-throughput siRNA knockdown study for 18,045 genes of the human genome performed in duplicate, using 4–5 unique pooled siRNAs per gene. After siRNA transfection, ciliation was induced by serum starvation, then fixed and imaged in 384-well plates in three channels (see 'Materials and methods'). 18 non-overlapping cellular features reflecting nuclear, cytoplasm and ciliary state, combined into 31 parameters (*Supplementary file 1*), yielding 559,395 values across the screen (*Supplementary file 2*).

## Development of the CILIOGENESIS data set

The rationale of our whole-genome siRNA screen with SEMG cells was to obtain data allowing for identification of genes as potential candidates as a cause of JS by using a supervised learning approach. We trained a Random Forest classifier using known 'ciliary genes' as a positive training set, derived from the SYSCILIA consortium gold standard (SCGSv1) composed of 303 confirmed factors (*van Dam et al., 2013*). The negative set incorporated genes not involved in any currently known ciliary processes and included 5445 genes annotated in the human metabolome database (HMDB 3.0) (*Wishart et al., 2013*), as well as a manually curated set of 666 housekeeping genes. To ensure accurate annotation of gene sets used in the classifier training, all genes were cross checked with Cildb V3.0, a database of 'ciliary genes' (i.e., genes with presumed ciliary function) based on high-throughput studies across multiple species (*Arnaiz et al., 2009, 2014*). Based on this resource, we removed genes with conflicting annotation from both the positive and negative sets, leaving a final list of high-confidence positive (n = 244) and negative (n = 1802) cilia candidates.

We evaluate the performance of the trained classifier on cilia candidates from the SCGSv1, which included an additional list of 419 ciliary gene candidates, not used to train the classifier. Of these, 21% were flagged by the classifier as likely ciliary. Furthermore, there was significant enrichment compared to the negative set of metabolomics and housekeeping genes not included in the classifier training set (p < 1.08 × 10$^{-25}$, one-tailed Wilcoxon rank sum).

Next, classifier performance was evaluated by examination of the area under the receiver operating characteristic-curve (AUC). Along with both replicates of the whole-genome siRNA screen, we included data from a siRNA screen designed to identify regulators of centriole biogenesis (*Balestra et al., 2013*) and gene expression signatures derived from the Genotype-Tissue expression (GTEx) tissue specific RNAseq data (*Figure 1D,E*, *Figure 1—figure supplement 1*, *Figure 1—figure supplement 2*) (*GTEx Consortium, 2013*). Of the 16,431 genes screened in all three data sets, the classifier predicted 1299 genes (7.9%) as likely ciliary, which we call the CILIOGENESIS database (Ciliary List of Candidate Genes using an siRNA Strategy, *Supplementary file 3A,B*). We also define a high-confidence subset of 591 ciliary genes by controlling for the false discovery rate (FDR < 0.1), which is estimated based on the classifier score and training set labels calculated. This high-confidence list includes many established ciliopathy genes such as *TTC26*, *CEP83*, *IFT88*, and *SPATA7*, as well as 14 of 25 known JS causative genes. Of the remaining JS causative genes, two others were included when FDR scores were loosened to 0.21 and 0.25. The remaining eight other JS causative genes (32%) were all found well above the genome-wide median classifier score (lowest ranked gene observed at 58th percentile), but not in the top list, possibly as a result of their activity

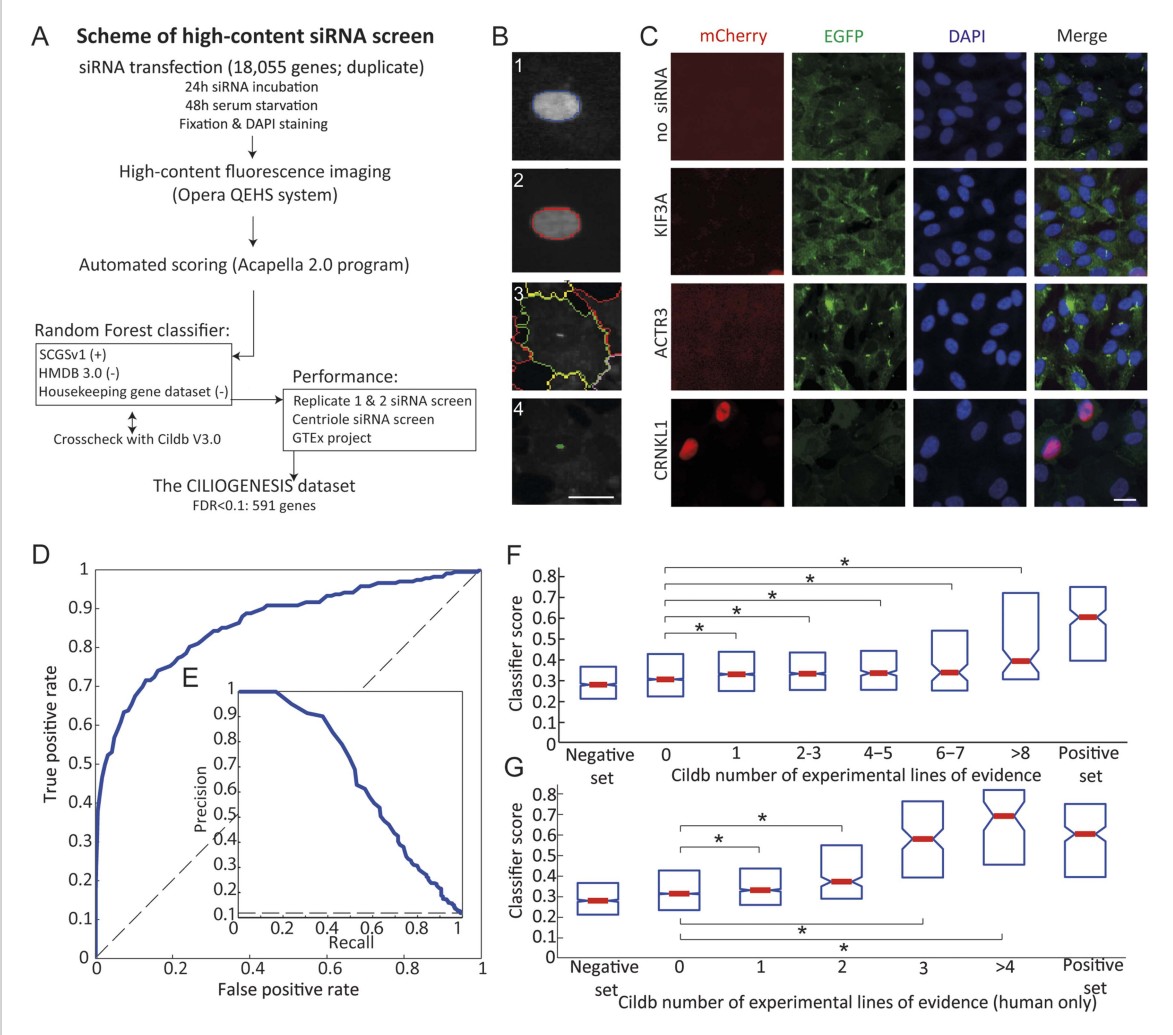

**Figure 1**. Schematic representation, validation and enrichment of genome-wide siRNA cell screen for machine learning approach. (**A**) High-content small interfering RNA (siRNA) cell-based screen using reverse transfection of the library in media containing serum for 72 hr, followed by 24 hr serum starvation, fixation and DAPI staining. Subsequent fluorescent imaging and algorithmic analysis performed for all pooled siRNAs. To assess ciliary candidates for the positive training, we used SYSCILIA gold standard (SCGSv1) and for the negative training the human metabolome database (HMDB 3.0) as well as a manually curated housekeeping gene data set. FDR, false discovery rate. (**B**) Segmentation algorithm for cytoplasm and cilia detection: (1) detected nuclei from DAPI channel, (2) nuclear automated segmentation, (3) cell outline automated using cytoplasm_detection_D of the program Acapella, and (4) cilia automated detection and segmentation. Images have been modified for illustration purposes. Scale bar: 10 μm. (**C**) Representative images of serum-starved SEMG cells without siRNA showing basal ciliation (small green rods in EGFP channel). Red (mCherry) marks cells in S/G2/M phase of the cycle, green (EGFP) marks cilia, blue (DAPI) marks nuclei. siRNAs used as positive controls: KIF3A interferes with ciliation but not cell cycle. ACTR3 shows increased length of cilia (*Kim et al., 2010*). CRNKL1 implicated in cell cycle progression (*Zhang et al., 1991*) and showed increased mCherry nuclei and reduced ciliation. Scale bar: 10 μm. (**D**) Receiver operating characteristic (ROC) for the classifier, which used features from three data sources. Dashed line: theoretical random classifier. (**E**) Precision-recall curve for the final classifier. (**F**) Median value (red center bar) and interquartile ranges (blue box) box plot of the classifier scores for the corresponding number of supporting number of evidences (NOEs) in Cildb and the genes used as negative and positive training examples. The indicated contrasts were found significant(*) with a highest value of p < 1.03 × 10⁻⁴ (one-tailed Wilcoxon's Rank sum test). (**G**) Same as (**F**), limited to the NOEs from humans only. The indicated contrasts were found significant(*) with a highest value of p < 1.43 × 10⁻¹⁰ (one-tailed Wilcoxon's Rank sum test). See *Figure 1—figure supplement 1, 2* for the prediction score on the gold standard and candidates as well as the visible improvement of the ROC curve and precision–recall curve.

The following figure supplements are available for figure 1:

**Figure supplement 1**. Prediction score on Gold standard and Gold standard candidates.

**Figure supplement 2**. Visible improvement of ROC curve and precision-recall curve.

outside the cilium. Of the high-confidence genes included in the CILIOGENESIS database, 26% were previously included in the SCGCv1, yielding 438 novel candidates.

Cildb is a multispecies knowledge base constructed through integration of high-throughput screens aimed at identifying ciliary or ciliary-related genes. Cildb outputs two integers for each gene in the knowledge base, referring to independent experimental 'number of evidences' (NOEs, i.e., publications) indicating ciliary association, with one for NOE in human studies and one for NOE in 'any species'. We compared gene-specific classifier score (excluding any genes used in training) with the Cildb NOE output. Significant positive trends were observed when comparing to increasing NOE in both the multi-species and human-only sets (Jonckheere–Terpstra test, see methods, $p < 3.04 \times 10^{-29}$ and $p < 6.50 \times 10^{-42}$, *Figure 1F,G*). Moreover, we also observed a significant difference when comparing scores in any of the NOE bins to the zero NOE bin in both the multi-species and human sets ($p < 1.03 \times 10^{-4}$ and $p < 1.43 \times 10^{-10}$, respectively, one-tailed Wilcoxon rank sum).

## Enrichment analysis of the CILIOGENESIS data set

To identify possible candidates for ciliopathies, we performed a gene ontology (GO)-term enrichment analysis on the high-confidence gene list, with functional annotation clustering using DAVID (*Huang da et al., 2009a*; *Huang da et al., 2009b*). We used a GO-enrichment cutoff of FDR <0.05 (Benjamini–Hocheberg test). To ascertain the novelty of genes included in the CILIOGENESIS data set, we excluded SCGSv1 genes used in the training, leaving 1,177 genes. GO enrichment resulted in several significant terms including non-membrane bound organelle, microtubule cytoskeleton/ centrosome, spermatogenesis, and microtubule cytoskeleton organization demonstrating an agreement with previous annotations for cilia associations (*Supplementary file 3C*). The involvement of ciliary processes in the CILIOGENESIS data set was supported by MsigDB analysis showing gene enrichment among others for the recruitment of mitotic centrosome proteins and complexes, microtubule/cytoskeleton and centrosome (*Supplementary file 3D,E*) (*Subramanian et al., 2005*). Enrichment validation suggested that the CILIOGENESIS data set may be enriched for ciliopathy disease genes.

## Intersection of CILIOGENESIS with unsolved JS cases highlights KIAA0586

Previous whole-exome sequencing in 287 cases of JS left ~50% without a genetic explanation (*Akizu et al., 2014*), suggesting additional causes remain to be identified. Of these, 75% displayed parental consanguinity, suggesting that causative variants might be homozygous. In about half of the remaining cases, sequencing on at least one parent was available, enabling phasing of identified alleles. From these 145 individuals, we tabulated 5485 variants containing 2348 homozygous variants and 3137 potentially compounds heterozygous variant pairs. We prioritized variants occurring within the coding region and canonical splice sites of any of the 591 CILIOGENESIS genes, and identified 179 variants including 106 homozygous and 73 potentially compound heterozygous variant pairs, or a 96.7% reduction in variants to be considered. Collectively, variants were identified in 112 of the 591 CILIOGENESIS genes, respectively. The only gene with more than two families displaying variants was *KIAA0586*, prompting further analysis.

*KIAA0586* (i.e., the orthologue of chicken and mouse *Talpid3*) is composed of 34 exons with at least six major transcripts (*Figure 2A*). From these 145 sequenced probands (written informed consent provided), there were four displaying putative compound heterozygous and two displaying homozygous potentially deleterious variants. Interestingly, in each of the four compound heterozygous probands, there was a shared frameshift mutation, (chr14:58899157del; c.428del, p. Arg143Lysfs*4), which we refer to as M1 (mutation 1). Each of the four carried a single additional potentially deleterious variant, including mutations in a canonical acceptor splice site (chr14: 58915212G>A; c.1120+1G>A, p.Thr323Hisfs*3; M2), a canonical donor splice site (chr14: 58923419G>C; c.1413-1G>C; p.Phe472Alafs*5; M3), and a missense affecting the start codon of two transcripts (chr14:58896138T>C; c.293T>C; p.Met98Thr; M4; T1; or c.2T>C; p.Met1?; T4-T5, where T refers to transcript number). Implementing an algorithm to identify copy number variants from exome-sequencing data (*Fromer et al., 2012*), we additionally identified a deletion of 15.5 kilobases (Kb) spanning exon 10–17 (chr14:?_58923420_58938997_?del; c.1413-?_2793+?del; p.?; M5) in one patient. These mutations were all confirmed with Sanger sequencing or quantitative PCR,

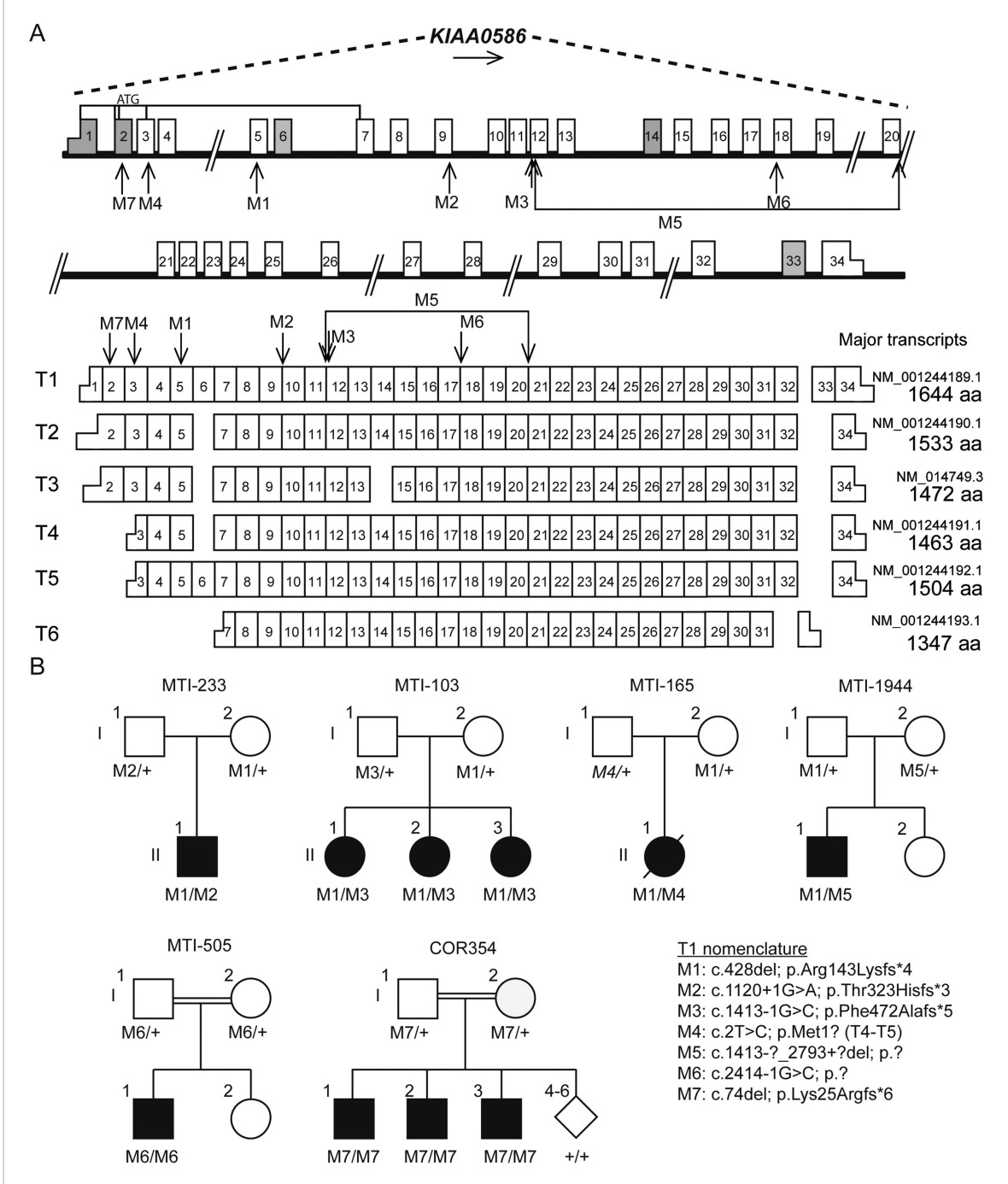

**Figure 2**. Pedigrees and schematic representation of *KIAA0586*. (**A**) Genomic structure and mRNA transcripts of *KIAA0586*. Transcript 1 (T1): full-length isoform with 34 exons. T2–T4 have different initiation sites, lack exon 5, and T3 lacks exon 14. T5 starts at the same position as T4 and incorporates exon 6. The shortest transcript (T6) initiates in exon 7, lacks exon 32 and 33, and terminates using an alternative exon, which is not incorporated in the other transcripts. Gray boxes represent alternative exons. UTR's are represented by half-height boxes. The location of the mutations is indicated by M1–M7. (**B**) Pedigrees of the Joubert syndrome (JS) families with ancestries of USA (MTI-233 and MTI-103), Mexico (MTI-165), Turkey (MTI-1944 and COR354), and Syria (MTI-505), respectively, demonstrating the segregation of the compound heterozygous mutations in non-consanguineous families and homozygous mutations in consanguineous families. Inferred genotype is italicized. M, mutation; T, transcript. See *Figure 2—figure supplement 1* for the chromatograms of the mutations in *KIAA0586*. *Figure 2—figure supplement 2* shows the results of the quantitative PCR confirming the large heterozygous mutation in MTI-1944.

The following figure supplements are available for figure 2:

*Figure 2. continued on next page*

*Figure 2. Continued*

**Figure supplement 1**. Chromatograms of mutations in the *KIAA0586* gene.

**Figure supplement 2**. Quantitative PCR confirmed heterozygous mutation in MTI-1944.

and all segregated according a strict recessive mode of inheritance in all available family members (*Figure 2B*, *Figure 2—figure supplement 1*, *Figure 2—figure supplement 2*). We conclude that compound heterozygous variants in *KIAA0586* contribute to JS. Each patient carrying the M1 mutation had a demonstrable second mutation on the other allele, suggesting a recessive mode of inheritance.

Two consanguineous families each showed a homozygous mutation in *KIAA0586*. One was predicted to alter splicing in a constitutively incorporated exon (c.2414-1G>C; p.?; M6). The other was a single base-pair deletion (c.74del; p.Lys25Argfs*6; M7), in an exon incorporated into only three of the six annotated transcripts, all of which are ubiquitously expressed. We conclude that homozygous mutations in *KIAA0586* can also contribute to JS.

The common frameshift variant M1 was identified in all four families with compound heterozygous mutations. Evaluation of M1 in the Exome Variant Server (NHLBI GO Exome Sequencing Project (ESP), Seattle, WA, URL: http://evs.gs.washington.edu/EVS/ [May, 2015]) identified in 25/7,757 European American alleles and 3/3511 African American alleles, all in a heterozygous state, presumably all in healthy individuals. Exome Aggregation Consortium (ExAC, Cambridge, MA, URL: http://exac.broadinstitute.org [May, 2015]) showed an overall frequency of 244/120,680 M1 alleles. Combining these with the 1000 Genomes data suggests an allele frequency of 0.0036 in the general population. We conclude that M1 is a relatively common allele in the general population, found in about 1/300 individuals. The M1 variant was found in individual of varying ancestry, but we cannot exclude a common founder mutation.

## Evaluation of KIAA0586 as a candidate gene in other JS cohorts

We speculated that M1 was likely to represent a common mutation among JS patients. Thus, we screened an additional cohort of 163 classical JS patients with a proven 'molar tooth sign' collected primarily from Mediterranean regions. The M1 allele was surprisingly identified in 17 of 326 alleles (5.21%), of which one was homozygous (*Figure 3* individual NG2872). Ethnically matched Mediterranean controls showed 2/536 M1 alleles (0.37%, $p < 0.0001$, odds ratio 13.51). In the remaining 15 individuals, we attempted comprehensive Sanger sequencing of the entire *KIAA0586* transcript, eventually identifying a pathogenic variant in eight individuals (57%), all leading to predicted splice, stop or frameshift changes, again consistent with recessive inheritance. In the other seven JS patients, a second mutation was not yet identified (*Table 1*, *Table 1—source data 1*). Although it is possible that one or more of these individuals carries M1 by chance, it is most likely that a second mutation exists, not yet uncovered.

To evaluate the effect of predicted splicing mutations in *KIAA0586*, we generated mRNA from cultured fibroblasts of an affected and unaffected member of family MTI-233 and MTI-103, displaying an M1 compounded with a splice mutation (M2 or M3, respectively). Sanger sequencing of poly-A primed mRNA showed that the mutation M2 led to the skipping of exon 9 and mutation M3 led to utilization of a cryptic splice acceptor located 16 bp downstream (i.e., 3′), resulting in a frameshifted transcript (*Figure 2—figure supplement 1B,C*), suggesting partial or complete loss-of-function.

Loss-of-function mutations in *Talpid3* result in a short-rib polydactyly-like phenotype in chicken and mouse, with a vascular defect and early lethality, all attributable due to defective ciliogenesis (*Bangs et al., 2011*; *Davey et al., 2014*). Our patients presented classical features of JS including the MTI of varying severity (*Figure 3*, *Figure 3—figure supplement 1*), without lethality or demonstrable excessive fetal wasting in affected families. Most cases displayed hypotonia, ataxia, developmental delay, and intellectual disability without skeletal or limb malformations. Breathing abnormalities, seizures, macrocephaly, and ophthalmological defects were found in a subset of the cases (*Supplementary file 4A*). The affected child of MTI-165 passed away at the age of 18 months from

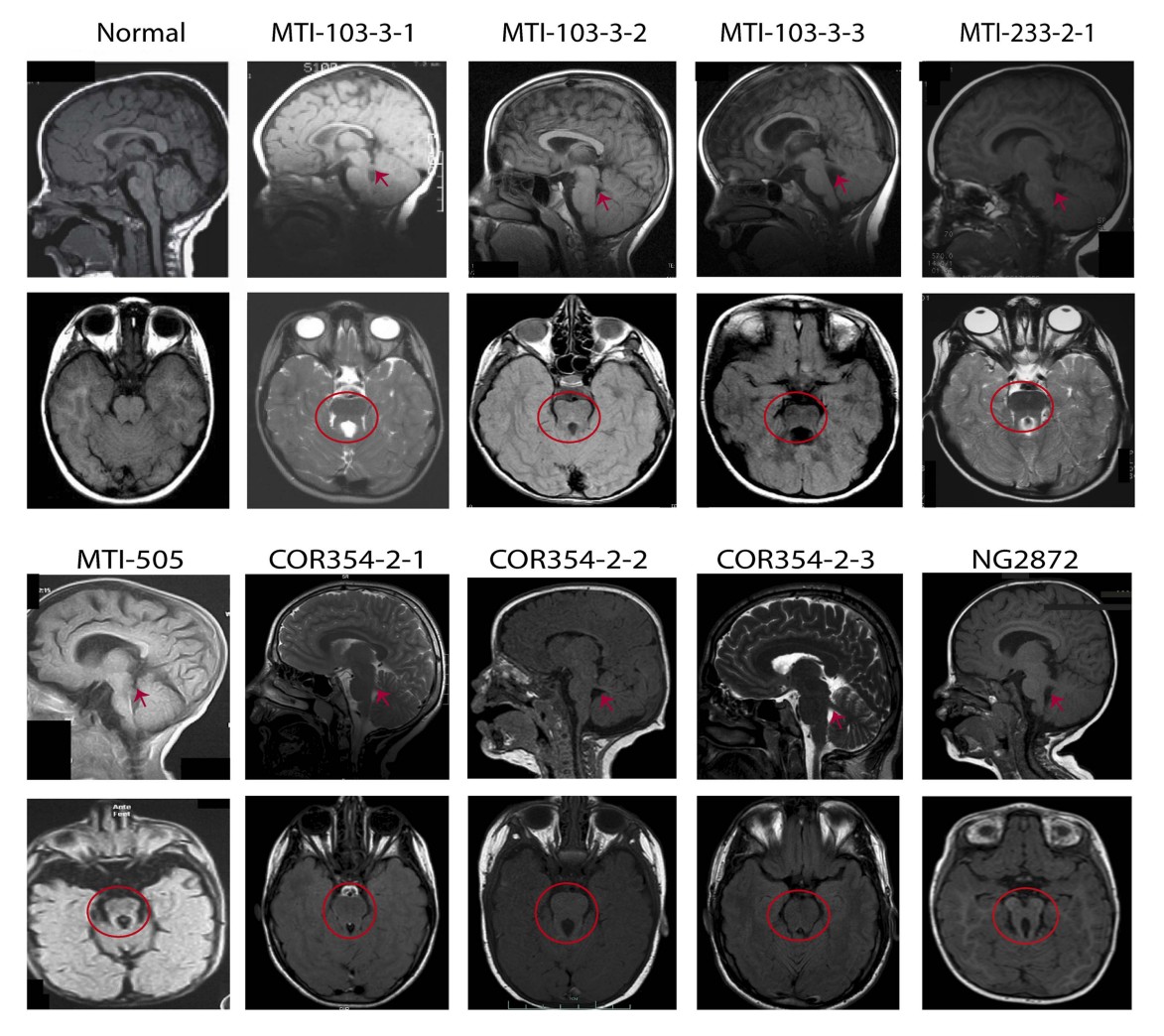

**Figure 3**. MRI scans from patients with *KIAA0586* mutations. Magnetic resonance imaging (MRI) in a healthy individual and patients with *KIAA0586* mutations showing thickened and mal-oriented superior cerebellar peduncle (upper, red 'arrowheads'), deepened interpeduncular fossa and constituting the 'molar tooth sign' (red circle). In COR-354-2-3, the molar tooth sign was very mild, possibly due to suboptimal image averaging. *Figure 3—figure supplement 1* shows the imaging phenotype of affected JS individual MTI-1944-2-1.

The following figure supplement is available for figure 3:

**Figure supplement 1**. Imaging phenotype of affected JS individual MTI-1944-2-1 with *KIAA0586* mutations.

apnea, and no imaging was available. The results support the involvement of *KIAA0586* in the pathogenesis of JS.

## Mutated KIAA0586 results in absence of detectable protein in patient cells

RT-PCR analysis with primers spanning various transcripts showed ubiquitous *KIAA0586* expression in various tissues (*Figure 4—figure supplement 1A*). To determine the effect of mutations on KIAA0586 protein level, we analyzed patient fibroblasts of family MTI-103 and MTI-233 by Western analysis using a KIAA0586-specific antibody (*Kobayashi et al., 2014*). The level of KIAA0586 protein in patient samples was below detection, whereas both carriers showed reduced but detectable expression compared with control (*Figure 4*). In human RPE1 cells transfected with KIAA0586 siRNA, we documented reduced protein levels, supporting antibody specificity.

**Table 1**. All alleles identified in *KIAA0586* causative for Joubert syndrome

| Patient ID | Genotype | Allele 1 (based on T1) | | | Allele 2 (based on T1) | | |
|---|---|---|---|---|---|---|---|
| | | Genomic | DNA | Protein | Genomic | DNA | Protein |
| MTI-233 | M1/M2 | g.58899157del | c.428del | p.Arg143Lysfs*4 | g.58915212G>A | c.1120+1G>A | p.Thr323Hisfs*3 |
| MTI-103 | M1/M3 | g.58899157del | c.428del | p.Arg143Lysfs*4 | g.58923419G>C | c.1413-1G>C | p.Arg472Serfs*2 |
| MTI-165 | M1/M4 | g.58899157del | c.428del | p.Arg143Lysfs*4 | g.58896138T>C | c.2T>C (based on T4-T5) | p.Met1? (based on T4-T5) |
| MTI-1944 | M1/M5 | g.58899157del | c.428del | p.Arg143Lysfs*4 | g.?_58923420_58938997_?del | c.1413-?_2793+?del | p.? |
| MTI-505 | M6/M6 | g.58934452G>C | c.2414-1G>C | p.? | g.58934452G>C | c.2414-1G>C | p.? |
| COR354 | M7/M7 | g.58895020del | c.74del | p.Lys25Argfs*6 | g.58895020del | c.74del | p.Lys25Argfs*6 |
| Mediterranean cohort analysis | | | | | | | |
| NG2872 | M1/M1 | g.58899157del | c.428del | p.Arg143Lysfs*4 | g.58899157del | c.428del | p.Arg143Lysfs*4 |
| NG4158 | M1/M8 | g.58899157del | c.428del | p.Arg143Lysfs*4 | g.58909503C>T | c.649C>T | p.Gln217* |
| NG2326 | M1/M9 | g.58899157del | c.428del | p.Arg143Lysfs*4 | g.58910790_58910791del | c.863_864del | p.Gln288Argfs*7 |
| NG1776 | M1/M9 | g.58899157del | c.428del | p.Arg143Lysfs*4 | g.58910790_58910791del | c.863_864del | p.Gln288Argfs*7 |
| NG3928 | M1/M10 | g.58899157del | c.428del | p.Arg143Lysfs*4 | g.58915097C>T | c.1006C>T | p.Gln336* |
| NG2458 | M1/M11 | g.58899157del | c.428del | p.Arg143Lysfs*4 | g.58924613_58924616delinsAAA | c.1658_1661delinsAAA | p.Val553Glufs*79 |
| NG2286 | M1/M12 | g.58899157del | c.428del | p.Arg143Lysfs*4 | g.58925263G>A | c.1815G>A | p.= / p.? |
| NG1485 | M1/M13 | g.58899157del | c.428del | p.Arg143Lysfs*4 | g.58927869C>T | c.2209C>T | p.Arg737* |
| NG3758 | M1/M14 | g.58899157del | c.428del | p.Arg143Lysfs*4 | g.58953883del | c.3462del | p.Gly1155Glufs*40 |

M; mutation; T; transcript. **Table 1—Source data 1** shows chromatograms belonging to the identified mutations in the Mediterranean cohort.

**Source data 1**. Chromatograms of mutations in the *KIAA0586* gene identified in the additional cohort of Mediterranean individuals with Joubert syndrome.

## Discussion

Here, we identify *KIAA0586* mutations in JS using a combination of cell-based screening and exome sequencing. By training of a classifier to prioritize ciliary candidate genes based upon shared loss-of-function phenotypes, we generated a data set we called CILIOGENESIS consisting of 591 prioritized genes. Intersecting these genes with WES data of genetically unexplained JS individuals led to the discovery of mutations in *KIAA0586*, which we found to be a relatively common cause (i.e., about 5%) in unsolved JS cases. In patient cells, there was undetectable KIAA0586 protein supporting its role in JS pathogenesis. It remains to be determined whether mutations in *KIAA0586* can lead to other ciliopathies like Meckel–Gruber syndrome or nephronophthisis, which are often allelic to JS.

Our siRNA screen incorporated several improvements over previously published but similar screens. As the first genome-wide siRNA high-content screen for defective ciliogenesis, we evaluated nearly each of the annotated human genes with at least four siRNAs per gene. Second, we incorporated a specific cell phase marker, mCherry-Geminin, to exclude false-positives that might result from cell cycle defects. Third, we incorporated a machine learning approach with positive and negative training sets, which enhanced the predictability of measured cellular features as they relate to ciliogenesis.

It is noteworthy that including features in the classifier from multiple sources, while improving performance of the classifier, caused a reduction from 18,045 targets to 16,431 targets due to missing values (n = 798 targets lost by the biogenesis siRNA screen; n = 786 targets lost by the GTEx RNAseq data). It is possible that some ciliary factors were not correctly classified as such due to incomplete data in these comparative screens. Inevitability, our machine learning approach will be biased towards currently known ciliary factors, and as more knowledge is gained, the power of such approaches will improve. Even by combining the CILIOGENESIS data set with exomes from 145

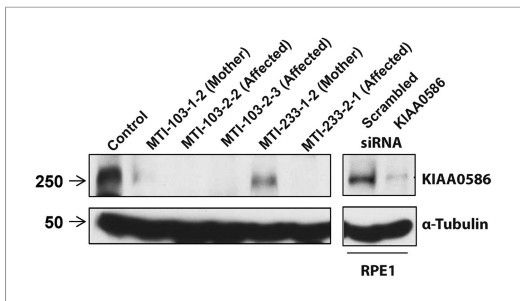

**Figure 4**. Absent KIAA0586 protein in patient fibroblasts. Immunoblot analysis of KIAA0586 in fibroblasts from family MTI-103 and MTI-233. Lysates from RPE1 cells transfected with scrambled or KIAA0586 siRNA were used as control. M, unaffected carrier (mother); A, affected child. RPE1, retinal pigment epithelial-1 cell line. *Figure 4—figure supplement 1A* represents an expression analysis of the *KIAA0586* gene.
The following figure supplement is available for figure 4:

**Figure supplement 1**. Expression analysis of the *KIAA0586* gene.

individuals identified only a single recurrently mutated gene, leaving the majority of families still unexplained (*Akizu et al., 2014*). This observation leads us to postulate that there are probably few commonly mutated genes remaining to be discovered in JS.

Our siRNA screen is probably underpowered to detect JS genes primarily involved in effects like signaling through Sonic hedgehog or Wnt pathways. Gene set enrichment analysis of the true positive SCGCv1 genes (SCGCv1 genes ranked within the CILIOGENESIS data set genes) with MsigDB (*Subramanian et al., 2005*) showed enrichment for cytoskeletal genes as expected (*Supplementary file 3F*), whereas analysis on false negative genes (SCGCv1 genes ranked outside the CILIOGENESIS data set genes) showed significant enrichment for photoreceptor cell maintenance, sensory perception, Sonic hedgehog pathway, and post-chaperonin tubulin folding pathway (*Supplementary file 3G*). This suggests that the CILIOGENESIS data set may be enriched for genes involved in the process of ciliogenesis, whereas genes involved in signaling functions are less likely to be detected. Moreover, this is in agreement with analysis of candidate targets involved in Hedgehog signaling from screens described in literature for which we observe no enrichment in the CILIOGENESIS data set (*Supplementary file 3A,B*) (*Evangelista et al., 2008*; *Jacob et al., 2011*). It is possible that extending the CILIOGENESIS data set to include factors regulating the ciliary responsiveness to Hedgehog or Wnt activators or suppressors could further improve sensitivity.

Talpid3 participates in the earliest stages of ciliation, including centriolar satellite dispersal and plasma membrane docking of the basal body (*Davey et al., 2014*; *Kobayashi et al., 2014*). Although *Cep290* and *Talpid3* share some similarities in ciliary phenotypes, there are distinct cellular functions (*Kobayashi et al., 2014*). Talpid3 forms a ring-like structure at the distal end of both centrioles and is involved in the initiation of ciliary vesicle formation and docking, whereas Cep290 functions in the maturation of these vesicles. Moreover, Talpid3 is localized asymmetrically in mother and daughter centrioles and is crucial for limiting the levels of Cep120 at the mother centriole (*Wu et al., 2014*). In *Talpid3* mutant mouse embryos, centrosomes fail to dock at the plasma membrane and cilia are absent in various tissues (*Yin et al., 2009*), associated with embryonic lethality.

*KIAA0586* might have been identified as mutated in JS even without the CILIOGENESIS data set, but was missed, probably for several reasons. First, the difference in names of the human and mouse genes made it difficult to link the two in automated curation of exome variants. Second, the majority of mutations were compound heterozygous, precluding homozygosity mapping analysis. Third, the higher frequency of the common allele M1 in the general population reduced its priority as a candidate, since the rarest alleles are prioritized over common alleles. Thus, we foresee the CILIOGENESIS data set and other orthogonal approaches as potentially beneficial in gene discovery.

The 1/300 calculated carrier frequency of M1 in the population is comparable to the deep intronic founder mutation of ~1/500 (c.2991+1655A>G) in *CEP290* as the most common cause of Leber congenital amaurosis in Caucasians, but less than the ~1/100 in *TMEM216* as a cause for JS in the Ashkenazi population (*den Hollander et al., 2006*; *Valente et al., 2010*). Of the 15 patients with heterozygous M1 in the Mediterranean cohort, we identified a second truncating allele in *KIAA0586* in 57%, and the remaining are still under investigation for non-coding or deletion mutations. We screened a cohort of 800 individuals with nephronophthisis with retinopathy, and found four carrying the M1 mutation, close to the predicted 0.0036 expected carrier frequency and no convincing second mutations were documented in this cohort. Thus, it remains to be determined if *KIAA0586* mutations are associated with other ciliopathy phenotypes or can lead to embryonic lethality.

Because the mutations affect only exons incorporated in a subset of transcripts or affect splicing (which can be leaky) and because of embryonic lethality in mouse and chick with homozygous null mutations, we speculate that humans surviving with *KIAA0586* mutations may retain partial function. The M4 allele was predicted to cause loss of the initiator methionine in transcript T4 and T5, potentially leaving other transcripts intact. M4 was encountered in public sequence databases ESP and ExAC with a frequency of 0.002 (322/132,340 alleles), including three homozygous cases with unknown health status. The M7 allele affects three of six transcripts, while no protein was detected on Western blot from patient cells. It will be important to model these alleles or check for complementation of two null alleles with the patient alleles.

## Materials and methods

### Cell culture
hTERT-transformed RPE1 cells were cultured in DMEM/F12 medium supplemented with 10% fetal bovine serum (FBS), under standard conditions (37°C, 5% CO$_2$). Plasmid DNAs harboring mouse Smo-EGFP and mCherry-Geminin (1–110aa) fusion genes were transfected to hTERT-RPE1 cells and the stable cell line; Smo-EGFP-mCherry-Geminin/hTERT-RPE1 (SEMG) was established by G418 selection. To induce ciliogenesis, the cells were serum starved on serum-free DMEM/F12 media for 24–48 hr prior to fixation.

### Whole-genome siRNA library screen

#### Primary screen
An arrayed library containing pooled siRNAs targeting 18,045 human genes (Dharmacon, Lafayette, CO) was screened in duplicate. Assay plates (384-well plate with optical bottom; Greiner Bio-One, Monroe, NC) were spotted with 1 µl of 0.5 µM siRNA using the Velocity 11-Bravo Pipette with a 384 ST head. Reverse transfection was performed using Lipofectamine RNAiMAX: final siRNA concentration was 10 nM. SEMG cells were suspended in DMEM/F12 supplemented with 10% FBS and seeded onto assay plates using the Matrix-Well Mate (2,000 cells in 40 µl medium for each well). Culture medium was replaced with DMEM 24 hr after transfection using the TiterTek-MAP-C, and cells were incubated for additional 48 hr before fixation in 4% PFA and subsequent staining with DAPI.

#### Imaging and image analysis
Image acquisition of the siRNA screen was performed on the Opera QEHS system (PerkinElmer, Waltham, MA). All cells were imaged with a 20× objective in a standardized manner using the Opera QEHS system (Perkin Elmer, Waltham, MA). The nuclei were stained with DAPI and exposed for ~10 ms using the non-confocal light path at 365-nm excitation with an and a 450/50-nm emission filter. The green fluorescence for expression of Smo was acquired at 488-nm excitation using the confocal system. The expression of Geminin was measured at 561-nm laser line using the confocal system. Each well was imaged in triplicate. Acapella 2.0 software (PerkinElmer, Waltham, MA) was used to perform image segmentation and cytometry with similar algorithms previously described (*Kim et al., 2010*). 31 output parameters were obtained by an algorithm generated for segmentation of the nucleus, cytoplasm, and primary cilium in the SEMG cells (*Figure 1A–C*, *Supplementary file 1*). The algorithm applied for segmentation of the nucleus, cytoplasm, and primary cilium in SEMG was confirmed by the manual imaging analysis in both serum positive and negative conditions.

#### Random Forest classification of cilia genes
Data generated by whole-genome siRNA high-content screen were quantile normalized across batches to facilitate cross validation. The SYSCILIA gold standard (SCGSv1) of known ciliary components (*van Dam et al., 2013*) was used as positive training examples. The SCGSv1 included 303 genes curated by the SYSCILIA consortium associated to a ciliopathy, ciliary localization, or function in ciliogenesis (*van Dam et al., 2013*). An additional list which included 419 candidate ciliopathy associated genes, which accompanied the gold standard, was used to benchmark the performance of our classifier and was excluded from training. As non-ciliary examples, we used two non-ciliary sets, the metabolome consisting of 5,445 genes (*Wishart et al., 2013*) and a manually created list of housekeeping genes of 666 genes. To further hone the positive and negative training sets, we use Cildb (V3.0) a comprehensive resource aggregating experimental evidence from 15 model organisms including humans (*Arnaiz et al., 2009*, *2014*). Genes

appearing in the Cildb list with any evidence of involvement in ciliary related processes were excluded (n = 9,073) from our negative training set, and in similar ways, genes in the positive training set were removed if evidence of ciliary involved was not seen in Cildb. The final positive training set composed of 244 genes, whereas in the negative training sets 1,802 genes remain. To prioritize candidate genes for ciliopathies, a Random Forest classifier was trained to accurately classify positive from negative samples based on features from data generated by our whole-genome siRNA screen, data from centriole formation from Balestra et al., and patterns of gene expression signatures across tissue from the GTEx project (*GTEx Consortium, 2013*).

First, the classifier was trained on the first replicate data set of the whole-genome siRNA experiment and tested on the second replicate and vice versa where a modest AUC of 0.63 and 0.64 was observed. Combining the features from the two batches, the classifier reached an AUC of 0.65 in test set performance (*Figure 1—figure supplement 1A*, *Figure 1—figure supplement 2*). Next, the classifier was trained with additional features collected in a centriole siRNA screen, which was a whole-genome siRNA study, was designed to identify regulators of centriole biogenesis and provide background on cilia, flagella, and centrosome formation (*Balestra et al., 2013*). Centriole data were downloaded from http://centriolescreen.vital-it.ch, to aggregate the effects of multiple siRNA, we use the weighted median method as in the ATARiS approach (*Shao et al., 2013*), which improved the AUC to 0.70 (*Figure 1—figure supplement 1B*, *Figure 1—figure supplement 2*). Subsequently, the GTExs (*GTEx Consortium, 2013*) data, which enables evaluation between genetic variation and gene expression in post-mortem human tissues, were used. We excluded 80 samples with low-RNA quality scores (RIN < 0.6), leaving 2,788 RNAseq samples from 52 tissues for further analysis. Reads per kilobase per million (RPKM) scores are quantile normalized across all samples. Next, for each tissue separately, we calculate the median expression RPKM score and principle component gene loading values for a set of leading principle components chosen to capture 95% of the total variance in each tissue (2–7 principle component, median 4 per tissue). By including these expression features derived from the GTEx RNAseq data in the classifier, an improvement to an AUC of 0.86 was reached (*Figure 1—figure supplement 1C*, *Figure 1—figure supplement 2*).

Classification was performed using the Random Forest approach (*Breiman, 2001*); trees were grown from bootstrapped samples of genes selected with replacement such that the number of negative samples matches the number of positive ones (randomized under sampling) (*Seiffert et al., 2010*). In each iteration, the square root the number of features was used (*mtry*, as suggested by Brieman et al.). Each forest is comprised of 5,000 trees trained as above (*ntree*). All predicted scores reported throughout our analysis are based on out-of-bag prediction scores (i.e., Random-Forest cross-validation scores).

## Gene set functional annotation clustering with DAVID

Functional annotation clustering of the CILIOGENESIS data set was performed with the online web tool DAVID (*Huang da et al., 2009b*). A set of 591 high scoring genes from the final joined classifier are used for the analysis (FDR < 0.1). CILIOGENESIS was tested for enrichment of GO FAT, KEGG, and Reactome pathway categories using the medium stringency setting of DAVID. As a background set, we use all genes, which have a full feature sets in all three data sources (16,810 genes; *Supplementary file 3C*).

## Gene set enrichment analysis with MsigDB

Gene set enrichment was performed by comparison against a collection of gene sets selected from the MsigDB (v5.0) database (Hallmark set, GO set, KEGG set, and Reactome set) (*Subramanian et al., 2005*). As a background set, we used all genes, which have a full feature sets in all three data sources (16,431). Sets larger than 400 or smaller than 5 were excluded, and only sets with a minimal overlap of three genes were included from the tested list in the p-value calculation. Enrichment p-value was calculated using a hypergeometric test of enrichment, and are only sets with FDR <0.1 are reported (estimated with B&H procedure).

## Jonckheere–Terpstra test of trend

When considering any type of evidence, the trend is tested for each individual bin (0, 1, 2, 3, 4, 5, 6, 7, >8). For 'human only' evidence, the trend was tested for bins of (0, 1, 2, 3, >4) (*Bewick et al., 2004*).

## Genetic analysis

### Patient Recruitment

Families were recruited for study based upon the presentation of JS in at least one member of the family. This study was approved by the institutional review boards of the participating centers. All subjects provided written informed consent (including consent to publish) prior to participation in the study. Sampling of blood for this study was performed on the proband and all affected and unaffected available genetically informative siblings and parents consistent with IRB guidelines or for skin biopsies from the proband and one parent when available. All patients were evaluated directly by one of the co-authors with specialty training in neurology, child neurology and/or clinical genetics, and in accordance with local medical practices. Detailed pedigree information, symptomatology, detailed general and neurological evaluations, brain/spine imaging and electrodiagnostic workup were performed in all affected members as well as clinically suspected members of each family, along with videos documenting the neurological examination in most cases.

### Exome sequencing

We performed WES in 145 families with affected(s) displaying features consistent with JS. Blood was acquired from informed, consenting individuals according to institutional guidelines, and DNA extracted using established protocols. In solution, exome capture was performed using the SureSelect Human All Exome 50 Mb Kit (Agilent Technologies, Santa Clara, CA) with 150-bp paired-end read sequences generated on a HiSeq2000 (Illumina, San Diego, CA). Sequences were aligned to hg19 and variants identified through the GATK pipeline (*DePristo et al., 2011*). Variations were annotated with in-house software and the SeattleSeq server (*Dixon-Salazar et al., 2012*).

### Systematic whole exome data analysis and variant identification

Initially, we systematically filtered for segregating (when WES of family member was present) autosomal variants with a total allele frequency <1% in Exome Variant Server (EVS; version ESP6500SIV2). Furthermore, all variants (except frame shifts variants) had a combined annotation dependent depletion_phred score ≥10 (CADD) (*Kircher et al., 2014*). All possible single nucleotide variants CADD scores were downloaded and provide a score to prioritize functional, deleterious and pathogenic variants across many functional categories, effect sizes and genetic architectures was unmatched by any current single-annotation method. Frameshift variants were included with a GERP-score ≥4.0 (*Cooper et al., 2005*). Homozygous variants were filtered out when present in unaffected individuals from our in-house database (n = 1,081), and compound heterozygous variants were removed when both were present in unaffected individuals. After performance of this script, we focused on the gene set of 591 genes of FDR <0.1 by applying a filter on the previous analysis. Variants in *KIAA0586* were analyzed for pathogenesis on the six largest transcripts (*Supplementary file 4B*) and segregation with disease within family members by regular PCR reaction. Primers for variant analysis and whole-gene scanning were designed using Primer3 (http://biotools.umassmed.edu/bioapps/primer3_www.cgi) (*Supplementary file 5A*).

### mRNA and gDNA analysis by RT-PCR

Quantitative PCR on genomic DNA was performed to confirm a the large deletion of unknown specific boundaries in MTI-1944. By analyzing two primer sets outside the deletion spanning exon 12 to 20 and two primer sets within the deletion quantity of PCR product was analyzed. Quantitative PCRs were performed using the C1000 Touch Thermocycler (Bio-Rad, Hercules, CA) in 96 micro-well plates. All samples were run in triplicate using iTaq Universal SYBR Green Supermix (Bio-Rad, Hercules, CA) mastermix, exonic primers (*Supplementary file 5B*) and template DNA. Input of genomic DNA was normalized against internal control gene *GAPDH*.

Total RNA was isolated from cultured fibroblasts from affected individual MTI-233-2-1 and MTI-103-2-2 and unaffected MTI-233-1-2 and MTI-103-1-2 according to manufacturer's protocol (Invitrogen, Carlsbad, CA). Reverse transcription with SuperScript III First-Strand Synthesis System (Invitrogen, Carlsbad, CA) was performed on 1 µg of total RNA. RT-PCR experiments were performed using 2.5 µl cDNA with primers in exons 8 and 10 (M2) and 10 and 12/13 (M3; intron spanning) (*Supplementary file 5B*) (35 cycles) followed by Sanger sequencing using a 3730 ABI DNA Analyzer.

### RNAi

Synthetic siRNA oligonucleotides were obtained from Dharmacon. Transfection of siRNAs using Lipofectamine 2000 or Lipofectamine RNAiMAX (Invitrogen, Carlsbad, CA) was performed according to the manufacturer's

instructions. The 21-nucleotide siRNA sequence for the non-specific control was 5′-AATTCTCCGAACGTGT CACGT-3′. The 21-nucleotide siRNA sequence for human Talpid3 is 5′-CAAAGTTACCTACGTGTTATT-3′.

## Western blotting

Fibroblasts were grown in DMEM supplemented with 10% FBS, grown to confluence, and subsequently serum starved for 72 hr to induce cilium growth. Cells were lysed with ELB buffer (50 mM Hepes pH 7, 150 mM NaCl, 5 mM Ethylenediaminetetraacetic acid (EDTA)/pH 8, 0.1% NP-40, 1 mM Dithiothreitol (DTT) DTT, 0.5 mM 4- benzenesulfonyl fluoride hydrochloride (AEBSF), 2 µg/ml leupeptin, 2 µg/ml aprotinin, 10 mM NaF, 50 mM ß-glycerophosphate, and 10% glycerol) at 4°C for 30 min. 100 µg of lysate per sample in sample buffer was loaded on SDS-PAGE gels. Proteins were transferred to a polyvinylidene difluoride (PVDF) membrane (GE Healthcare, Little Chalfont, UK) and blocked in 3% non-fat milk in Phosphate-buffered saline (PBS). Rabbit polyclonal antibody against Talpid3 (dilution 1: 1,000) (Kobayashi et al., 2014) and a mouse monoclonal antibody against α-tubulin (Sigma–Aldrich, dilution 1:5,000) were incubated overnight at 4°C.

## Acknowledgements

We thank the affected children and their families for their invaluable contributions to this study, supported by National Institutes of Health grants (R01NS041537, R01NS048453, R01NS052455, P01HD070494, P30NS047101 to JG Gleeson; 1R01HD069647-03 to S Kim and BD Dynlacht and R01DK068308 to F Hildebrandt), the Howard Hughes Medical Institute, and Simons Foundation (JG Gleeson). We thank the Broad Institute (U54HG003067 to E Lander), the Yale Center for Mendelian Disorders (U54HG006504 to R Lifton and M Gunel) for sequencing support. This work was also partly supported by grants from the Italian Ministry of Health (Ricerca Corrente 2015 to EM Valente), the Telethon Foundation Italy (Grant GGP13146 to E Bertini and EM Valente), and the European Research Council (ERC Starting Grant 260888 to EM Valente).

## Additional information

### Competing interests

JGG: Reviewing editor, *eLife*. The other authors declare that no competing interests exist.

### Funding

| Funder | Grant reference | Author |
|---|---|---|
| National Institutes of Health (NIH) | R01NS041537 | Joseph G Gleeson |
| European Research Council (ERC) | 260888 | Enza Maria Valente |
| Howard Hughes Medical Institute (HHMI) | | Joseph G Gleeson |
| Simons Foundation (SF) | | Joseph G Gleeson |
| Fondazione Telethon (Telethon Foundation) | | Enza Maria Valente |
| National Institutes of Health (NIH) | 1R01HD069647-03 | Sehyun Kim, Brian D Dynlacht |
| National Institutes of Health (NIH) | P03NS047101 | Joseph G Gleeson |
| National Institutes of Health (NIH) | P01HD070494 | Joseph G Gleeson |
| National Institutes of Health (NIH) | R01NS052455 | Joseph G Gleeson |
| National Institutes of Health (NIH) | R01NS048453 | Joseph G Gleeson |
| National Institutes of Health (NIH) | R01DK068306 | Friedhelm Hildebrandt |

The funders had no role in study design, data collection and interpretation, or the decision to submit the work for publication.

## Author contributions

SR, MH, SK, JGG, Conception and design, Acquisition of data, Analysis and interpretation of data, Drafting or revising the article, Contributed unpublished essential data or reagents; ES, BC, JLS, ROR, JS, TM, EM, MSZ, KJS, JM-D, WBD, MAM, FI, MA, RB, RR, R-MB, CLC, SD'A, PS, EB, FS, MM-B, IM, EB, FE, MS, FH, MF, KKV, SBG, Acquisition of data, Analysis and interpretation of data; MR, CAJ, BDD, EMV, Acquisition of data, Analysis and interpretation of data, Drafting or revising the article; PA-B, SH-G, Conception and design, Acquisition of data, Analysis and interpretation of data; TI, JEL, JK, Conception and design, Acquisition of data

## Ethics

Human subjects: Consenting and sampling was performed on both parents and all available genetically informative siblings to include affected and unaffected members, as well as extended family members if appropriate, consistent with IRB guidelines approved by the ethical committee (JGE-0853) and according to the Declaration of Helsinki.

# Additional files

## Supplementary files

• Supplementary file 1. Parameter output genome-wide siRNA analysis screen. Table listing all parameters used in the analysis of the genome wide siRNA screen.

• Supplementary file 2. siRNA experimental output based on 31 parameters. Table listing all raw measurements based on the 31 parameters used in the analysis of the genome-wide siRNA screen in duplicate.

• Supplementary file 3. The CILIOGENESIS data set. (A) Whole genome table listing the rank of predicted ciliary genes and non-ciliary genes. (B) The CILIOGENESIS data set containing the genes predicted by the classifier to be ciliary. (A–B) Genes predicted by the classifier to be ciliary are color coded. Green rows represents the genes with an FDR <0.01, yellow with FDR <0.1, orange with FDR <0.2, and the remainder (until FDR <0.267) is colored in dark orange. (C) Enrichment analysis by performing enrichment analysis with DAVID on the CILIOGENESIS database (excluding SCGCv1 genes). (D) Enrichment analysis with MsigDB (v4.0, gene sets from GO, KEGG and Reactome) on the CILIOGENESIS database (excluding SCGCv1 genes). (E) Enrichment analysis by performing enrichment analysis with MsigDB on the genes with FDR <0.01 within the CILIOGENESIS database (excluding SCGCv1 genes). (F) Enrichment analysis of hypergeometric gene set on MsigDB onto the true positive genes (SCGCv1 genes ranked within the CILIOGENESIS data set genes). (G) Gene set enrichment on MsigDB onto false negative genes (SCGCv1 genes ranked outside the CILIOGENESIS data set genes).

• Supplementary file 4. Clinical features and *KIAA0586* mutations. (A) Clinical features of individuals with *KIAA0586* mutations. (B) Nomenclature per isoform of the identified *KIAA0586* mutations.

• Supplementary file 5. Primers. (A) Primers for *KIAA0586* mutation confirmation and segregation analysis. (B) Primers for *KIAA0586* mRNA and expression analysis.

## Major datasets

The following previously published datasets were used:

| Author(s) | Year | Dataset title | Dataset ID and/or URL | Database, license, and accessibility information |
|---|---|---|---|---|
| van Dam TJ, Wheway G, Slaats GG, SYSCILIA Study Group, Huynen MA, Giles RH | 2013 | The SYSCILIA gold standard (SCGSv1) | http://www.syscilia.org/goldstandard.shtml | Publicly available at SYSCILIA (Accession no: 23725226). |

| Author(s) | Year | Dataset title | Dataset ID and/or URL | Database, license, and accessibility information |
| --- | --- | --- | --- | --- |
| Wishart DS, Jewison T, Guo AC, Wilson M, Knox C, Liu Y, Djoumbou Y, Mandal R, Aziat F, Dong E, Bouatra S, Sinelnikov I, Arndt D, Xia J, Liu P, Yallou F, Bjorndahl T, Perez-Pineiro R, Eisner R, Allen F, Neveu V, Greiner R, Scalbert A | 2013 | HMDB 3.0–The Human Metabolome Database | www.hmdb.ca | Publicly available at Human Metabolome Database (Accession no: 23161693). |
| Balestra FR, Strnad P, Flückiger I, Gönczy P | 2013 | Centriole screen | http://centriolescreen.vital-it.ch | Publicly available at Centiole Screen (Accession no: 23769972). |
| Consortium | 2013 | The Genotype-Tissue Expression (GTEx) project | http://www.gtexportal.org/home/ | Publicly available at Genotype-Tissue Expression (Accession no: 23715323). |

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
