## [Decision Letter]

Thank you for sending your work entitled “A functional genome-wide siRNA screen identifies *KIAA0586* as mutated in Joubert syndrome” for consideration at *eLife*. Your article has been evaluated by Stylianos Antonarakis (Senior editor) and two reviewers, one of whom, Harry Dietz, is a member of our Board of Reviewing Editors.

Overall, the manuscript was very favorably reviewed. However, a number of points were raised that need to be addressed by reformatting or in the Discussion before final acceptance can be offered.

The Reviewing editor and the other reviewer discussed their comments before we reached this decision, and the Reviewing editor has assembled the following comments to help you prepare a revised submission.

Given the creative and seemingly focused nature of the primary functional screen, one is left wondering whether the application of the supervised learning approaches and training datasets has exacted too high a cost. It seems notable that these filters resulted in exclusion of many known JS causative genes (that were present in the 1,952 candidates) from the prioritized dataset (235 genes). In theory, these genes are indeed involved in ciliogenesis, but were excluded due to excessive filtering, a practice that might also limit or preclude the identification of genes with entirely novel ciliogenesis-related functions. It is also notable that many other JS causative genes were not represented in the broader (functionally-determined) dataset. While, in theory, this may relate to a contribution to ciliary function (as opposed to biogenesis), this is never specifically addressed. There should be a more comprehensive and detailed consideration of both the apparently false-negative and false-positive inclusions in both the broader and restricted CILIOGENESIS datasets. In this light, it is notable that the Discussion ends on a somewhat bland note. There should be more about limitations and strengths of the CILIOGENESIS dataset and anticipated efforts toward improvement.

---

## [Author Response]

*Given the creative and seemingly focused nature of the primary functional screen, one is left wondering whether the application of the supervised learning approaches and training datasets has exacted too high a cost. It seems notable that these filters resulted in exclusion of many known JS causative genes (that were present in the 1,952 candidates) from the prioritized dataset (235 genes). In theory, these genes are indeed involved in ciliogenesis, but were excluded due to excessive filtering, a practice that might also limit or preclude the identification of genes with entirely novel ciliogenesis-related functions. It is also notable that many other JS causative genes were not represented in the broader (functionally-determined) dataset. While, in theory, this may relate to a contribution to ciliary function (as opposed to biogenesis), this is never specifically addressed. There should be a more comprehensive and detailed consideration of both the apparently false-negative and false-positive inclusions in both the broader and restricted CILIOGENESIS datasets. In this light, it is notable that the Discussion ends on a somewhat bland note. There should be more about limitations and strengths of the CILIOGENESIS dataset and anticipated efforts toward improvement*.

We would like to thank the reviewers for this careful comment. To address this concern we have made concerted efforts at improving our overall classifier performance, which we are pleased to report proved extremely fruitful.

The improvement to our classifier is the result of a careful reanalysis of our feature processing approach. First we removed multiple normalization steps, which negatively impacted the classifier performance and were the result of choices made early in the classifier development process. These choices were appropriate when using supervised learning approaches, e.g. ‘support vector machines’, but were unnecessary and degraded performance when using random-forest a method based on scale invariant decision rules. We observed a further improvement in classifier precision by retaining tissue specific expression information from GTex. We excluded from analysis expression samples with low RNA quality score (RIN<6) leaving 2785 samples from 52 tissue types, including tissues from 10 different brain regions. Processing each of these tissues in the same manner as was originally applied globally to the entire set we extracted a variable number of features from each tissue in order to capture 95% of variance in each tissue (median 4 features). Finally, we included known JS candidate genes as positive examples in addition to the SCGSv1 set. Overall these changes resulted in a modest improvement of AUC (0.84 to 0.86), but a substantial improvement to classifier precision and FDR: The improved CILIOGENESIS dataset consists of fewer overall genes flagged (1299 vs. 1925), but a larger set (591 vs. 204) high confidence ciliary genes (FDR <0.1). This set includes 16 of the currently known JS genes, of which 5 are observed with a FDR<0.01, and 14 with FDR <0.1. Furthermore, the revised set now includes over 26% of SCGSv1 genes in this list (up from 7%). We note that in the previous version of our manuscript this number was incorrect.

Next, to better understand the limitation of the supervised analysis we examined the set of true-positive (TP) and false-negative (FN) genes based on this classifier (i.e. the classifier recall performance). The TP set of genes included all successfully detected examples, i.e. positive training samples (high confidence SCGCv1 and known JS) that were successfully detected by the classifier. The FN set included the remaining positive examples that the classifier was unable to detect. To characterize these sets we performed enrichment analysis of GO-terms as well as KEGG and Reactome pathways. We observed that in the TP set significant terms included microtubule organizing center, centrosome organization and biogenesis, microtubule motor activity, and gamete generation. Similar analysis on the FN set showed significant GO term enrichment for photoreceptor cell maintenance, sensory perception, Sonic hedgehog pathway and post-chaperonin tubulin folding pathway ([Supplementary-material SD4-data], sheet C and D). This suggested that the CILIOGENESIS dataset was better able to detect genes involved in the process of ciliogenesis, whereas genes involved in ciliary function such as signaling were not as well represented. Moreover, we performed an analysis of candidate targets involved in Hedgehog and WNT signaling from screens described in the literature and observed no statistically significant overlap with either the TP, FN or the CILIOGENESIS dataset ([Supplementary-material SD4-data], column O and P).

Using the newly created high confidence candidate set (FDR <0.1) we performed the same stringent analysis of the WES data. While we observed new CILIOGENESIS candidate genes in which unexplained JS patients have exome variants, we believe that the careful interrogation of these new candidates is beyond the scope of this manuscript. We are pleased that the revised process led to this improved methodology in flagging ciliary genes for the benefit of the field of ciliopathy research.